# SAGE: A Framework for Semantic-Alignment-Guided Engineering of Prompts and Fine-Tuning in Industrial Control Tasks

## Abstract

Large language models show great potential for code generation tasks, but automatic code generation for industrial control systems still faces challenges such as inaccurate semantic understanding, a lack of alignment evaluation, and a shortage of domain-specific fine-tuning models. Given the stringent requirements for real-time performance, security, logical rigor, and correct execution of industrial control code, existing general-purpose methods struggle to meet these demands. Therefore, this paper proposes a semantic alignment-guided prompt engineering approach for industrial control tasks. The approach consists of three core components: first, a dataset of function prompt formats covering five structured prompt patterns and a selection of 1,500 prompt examples for industrial control tasks is constructed; second, a semantic alignment analysis metric is designed to evaluate the semantic correctness and task consistency of code generated by different models; and third, an alignment-guided fine-tuning strategy is proposed, leveraging prompt-output-intent triples to enhance the model's generation capabilities for industrial control tasks. Experiments are conducted on five mainstream 7B models: DeepSeek-7B, Qwen2.5-7B, InternLM2-7B, Mistral-7B, and Gemma-7B. Results show that after fine-tuning, the executable performance of Mistral-7B and DeepSeek-7B increased from 0.719 to 0.886 and from 0.676 to 0.837, respectively, and the BLEU scores increased from 3.79 to 7.45 and from 3.45 to 6.62, respectively. All models maintained intent consistency (Intent = 1.000). Gemma-7B and Qwen2.5-7B showed decreases in executable performance, success rate, and BLEU, suggesting possible overfitting or distribution mismatch issues. The method proposed in this paper significantly improves the code executable performance and semantic alignment of some models in industrial control scenarios. It also reveals the sensitivity of model architecture to fine-tuning strategies, providing an important reference for subsequent architecture-aware alignment optimization.

## 1 Introduction

In recent years, generative artificial intelligence has developed rapidly and demonstrated its potential in many fields. Among them, code generation, as one of the important application areas of generative artificial intelligence, has received increasing attention. Code generation technology aims to reduce the workload of developers and improve programming efficiency by automatically generating source code that meets specific requirements.

The research on automatic code generation has a long history. In the early days, it mainly relied on compilers to directly map high-level languages into machine code. Although the conversion from languages such as Fortran and COBOL to assembly was achieved, it lacked optimization and portability and was difficult to adapt to complex application scenarios (Aho et al., 2006). Since 1990, modern compilers represented by GCC and LLVM have introduced cross-process optimization and multi-target backends, allowing the same optimization logic to be applied to multiple languages and platforms (Lattner & Adve, 2004). Around 2000, the model-driven development (MDD) method emerged, generating code through abstract models such as UML and SysML, and combining tools such as Simulink to achieve rapid prototyping of control systems (Schmidt, 2006). After 2010, it

has become possible to automatically generate interlocking, control logic, and other codes based on requirement rules or industry standards (such as IEC 61131-3). Although the degree of automation has increased, the generated results are often rigid and difficult to handle complex semantics and unstructured inputs (De Smet et al., 2008). With the emergence of large language models such as GPT-4, code generation has begun to move from "rule-driven" to "intention-driven." By understanding natural language instructions, the model can directly generate complete functions, control logic, and even cross-domain hybrid codes (Chen et al., 2021), effectively alleviating the problem that traditional modeling methods are difficult to cover complex semantics.

Unlike general code generation tasks, industrial control code generation requires not only functional correctness but also real-time performance, security, and standard compliance (Krüger et al., 2012). Its tasks involve ladder diagram instruction sets, PID parameter self-tuning, and industry standards such as IEC 61131-3 and ISO26262. In this context, although large language models have the ability to generate syntactically correct code, they still have obvious deficiencies in understanding the domain semantics of industrial control. For example, the emergency stop (ESD) logic of the PLC must strictly follow the timing constraint of "cutting off the output first and then triggering the alarm". If the generated logic is reversed, it may cause serious safety risks. At the same time, the "hallucination" problem common in large models is even more fatal in industrial control scenarios (Ouchani & Fakih, 2024). Fictional functions or communication rules have limited impact in general applications, but in control systems, they may cause motor overload and burnout or equipment communication failure, thereby inducing system-level accidents.

In response to the above challenges, some studies in recent years have explored the application of large language models to industrial control code generation tasks. For example, the AlphaCodium system proposed by Zhao et al. uses control flow tracking and rapid engineering mechanisms to enhance the model's ability to model temporal dependencies and function structures, alleviating the hallucination problem to a certain extent (Ouchani & Fakih, 2024). The LLM4PLC framework proposed by Fakih et al. combines semantic templates with formal verification mechanisms to generate interlocking and control logic programs that comply with the IEC61131-3 standard, effectively improving the reliability and standardization of the generated code (Fakih & Ouchani, 2024a). In addition, there are also studies that attempt to further improve the generation stability and interpretability of large models in industrial environments through constraint injection, template guidance, and other methods, and promote their application in actual engineering scenarios.

However, most of these methods remain at the experimental verification stage and lack a unified prompt design and evaluation system, making them difficult to meet actual industrial needs. Currently, key challenges facing the industrial control field include: first, the lack of a suitable, high-quality, and adaptable set of industrial prompts; second, the lack of methods to detect semantic alignment between these prompts and generated results; and third, the lack of fine-tuned industrial control models for evaluation and comparison. Therefore, there is an urgent need to build an industrial control system capable of prompt regulation, alignment detection, and model fine-tuning to support performance evaluation and optimization of basic models using metrics such as BLEU and Pass@k.

To address these issues, this paper proposes a semantic alignment guidance method for industrial control code generation. The contributions of this paper can be summarized as follows: It proposes a unified prompt design method, a semantic alignment evaluation mechanism, and an alignment-driven model fine-tuning strategy. Through techniques such as structured prompt construction, semantic label recognition, and alignment-guided training, it systematically addresses key issues such as a lack of prompt sets, difficulty in intent detection, and model mismatch, resulting in significantly improved generation results and reliable alignment of industrial semantics.

First, we constructed the Functional Framework Prompt (FFP) dataset. This dataset contains 1,500 prompt words across 10 categories, addressing the current lack of a suitable, high-quality set of prompt words. These prompt words cover various aspects of structured text content. This dataset provides essential information for subsequent evaluations such as alignment and fine-tuning of large language models, making it suitable for comparing and improving code generation performance.

Secondly, a Semantic Alignment Analysis (SAS) method was proposed to evaluate the alignment of code generation results with industrial control semantics. This method constructs a key semantic tag system to annotate and match the generated code to ensure that functions such as sensor reading,

actuator control, and interlocking logic are correctly implemented. This method quantifies the consistency of model output with the intended control intent. This method is based on a cross-language semantic taxonomy that covers key control intents such as flow control, temperature control, and interlocking logic. SAS quantifies the alignment of generated code with the intended functional categories, providing a unified evaluation framework for different programming languages.

Third, we propose an alignment-guided LoRa fine-tuning (AGFT) strategy. This strategy leverages semantic alignment scores to select and prioritize high-quality prompt-output pairs. Based on this, LoRa is used for efficient fine-tuning, enabling large language models (LLMs) to better adapt to industrial control tasks while maintaining generation safety and domain compliance. By constructing prompt-output-intent (POI) triples, AGFT enhances the model's ability to perceive control objectives and improves metrics such as code generation feasibility, task consistency, and success rate.

The remainder of this paper is organized as follows: "Related Work" outlines the basic preparations. "Proposed Method" presents the experimental workflow. "Function Prompt Format (FFP)" introduces our prompt word set and sample standard code set; "Semantic Alignment Analysis (SAS)" introduces an evaluation system for aligning prompts and answers in industrial control systems; and "Alignment-Guided Fine-Tuning (AGFT)" compares the performance of the LoRA fine-tuned model with the original model. "Experiments" presents the experimental platform and experimental results, followed by a discussion and analysis. "Conclusion" summarizes this paper and identifies any shortcomings and future work.

## 2 RELATED WORK

### 2.1 PROMPT ENGINEERING

Prompt engineering has become a key technology for adapting large language models (LLMs) to downstream tasks such as code generation. Zhou et al. (2024b) first proposed a standard for evaluating the reliability of LLM output from the perspective of trust and consistency, providing a reference for subsequent prompt design. Subsequently, Shin et al. (2020) compared the performance differences between prompt-based and fine-tuned models in code-related tasks, while Lester et al. (2021) proposed a parameter-efficient prompt fine-tuning method, providing a practical path for actual systems. In order to improve prompt efficiency, Zhou et al. (2024a) proposed a cost-effective and efficient prompt generation method based on search. At the theoretical and methodological level, Liu et al. (2023) systematically reviewed the prompt framework and proposed a classification of methods and applications; Mohanty et al. (2025) further extended it to adaptive prompts, proposed a framework driven by learnability, and emphasized risk control in multimodal environments. At the same time, Kojima et al. (2022) proposed a zero-shot reasoning prompt method, providing new ideas for practitioners and researchers. Furthermore, Gu et al. (2024) explored the role of prompt tags in enhancing instruction tuning performance, highlighting the importance of tag-level prompt design. Subsequently, Springer et al. (2024) compared various prompt engineering techniques through empirical research, revealing the effectiveness of prompt design in different task scenarios. Finally, Wang et al. (2024c) focused on the relationship between target code complexity and prompt strategies, emphasizing the necessity of complexity-aware prompts.

### 2.2 ALIGNMENT/EVALUATION METRICS

Although prompt engineering has improved the adaptability of LLM in code generation, ensuring that the generated results are aligned with human needs remains a key issue, which has led to the research on alignment and evaluation metrics. Aligning large language models (LLMs) with human values, preferences, and safety requirements has become a key research direction. At the evaluation level, Chiang et al. (2024) proposed a semantic relevance metric designed specifically for prompt evaluation to capture a deeper level of alignment between prompt representations and generated outputs, providing a useful supplement to traditional reward models or BLEU-type metrics. Subsequently, Shen et al. (2023) conducted a fundamental review of LLM alignment methods, focusing on supervised fine-tuning and reinforcement learning with human feedback (RLHF). On this basis, Wang et al. (2024b) proposed a more comprehensive taxonomy covering RLAIF and PPO-based methods and pointed out that there are still challenges in building a scalable and generalizable alignment framework. For multimodal scenarios, Yu et al. (2025) explored the adaptability of alignment

methods when combining language models with visual or audio modalities, emphasizing the complexity of preference alignment in cross-modal environments. Finally, Zhang et al. (2024b) proposed a roadmap for a scalable automatic alignment process, combining automated feedback collection and tuning loops, focusing on the scalability and stability of the evaluation.

## 2.3 COMMAND ADJUSTMENT/FINE-TUNING

As alignment methods continue to evolve, instruction tuning, as a core means to improve the performance of models in specific tasks, has gradually become a research focus. It has been proven that instruction tuning is an important strategy for adapting large language models (LLMs) to specific domain tasks, especially in code generation. Wei et al. (2021) first proposed the basic idea of instruction tuning and verified its effectiveness in code generation. Subsequently, He et al. (2024) proposed a secure code generation pipeline that uses an instruction tuning model to enforce security and compliance constraints. Zhong et al. (2024) presented an empirical method for transitioning from prompt engineering to full fine-tuning, with special attention to actual deployment scenarios and customized frameworks. As a supplement, Ma et al. (2024) proposed LLaMoCo, which optimizes code generation through instruction tuning to improve computational efficiency. In terms of comparing different methods, Wang et al. (2023) conducted an empirical comparison between instruction tuning and prompt engineering, while Sun et al. (2024) conceptually explained the core principles of instruction tuning and distinguished it from other tuning paradigms. Liu et al. (2024b) further explored a hybrid optimization strategy that combines fine-tuning with prompt engineering. Finally, Xu et al. (2023) conducted a comprehensive analysis of code-specific instruction tuning models and evaluated their performance on a variety of code-related tasks.

## 2.4 INDUSTRIAL CONTROL SYSTEMS

The above methods have made progress in general tasks, but in the high-reliability scenario of industrial control systems, they still need to be explored in combination with domain characteristics. Large language models (LLMs) have recently been widely explored in the field of industrial automation and control systems. Zhang et al. (2024a) pointed out that AI-generated code may bring critical supply chain risks in industrial environments, while Li et al. (2024) analyzed the internal mechanism of LLM in detail and systematically explained the operation and response behavior of the converter. Furthermore, Chen et al. (2025) explored the programming workflow based on LLM from a practitioner's perspective and emphasized potential problems in application scenarios. In terms of open source research, Wang et al. (2024a) compared retrieval-augmented generation (RAG), fine-tuning, and prompt engineering methods, and provided a benchmark for collaborative evaluation. In terms of framework exploration, Fakih & Ouchani (2024b) proposed LLM4PLC, a verifiable PLC programming framework based on LLM, which aims to address the correctness and safety constraints of control logic. Liu et al. (2024a) proposed an optimization framework that integrates prompt engineering, RAG, and fine-tuning, while Klein & Smith (2024) proposed a formal framework for "generative prompt engineering" to strengthen the theoretical foundation and measurement indicators. At the same time, Wei et al. (2022) proposed a Chain-of-Thought prompting framework and experimentally demonstrated the key role of prompt design in complex reasoning tasks. In terms of specific method research, Singh & Patel (2024) analyzed the interaction between fine-tuning and prompt design in LLM-based code generation and proposed a hybrid strategy suitable for engineering tasks; Zhao et al. (2024) proposed the AlphaCodium system, which combines prompt engineering with control flow tracing to improve code generation performance.

## 3 METHODOLOGY

Figure 1 shows the framework of the method proposed in this paper. The Functional Framework Prompt (FFP) dataset is proposed, which contains industrial control prompts and five prompt types. Semantic Alignment Analysis (SAS) is designed to test the quality of FFP using semantic labels and semantic alignment scores. Alignment-Guided Fine-Tuning (AGFT) is proposed to fine-tune models such as DeepSeek-7B and Qwen2.5-7B using the Prompt-Output-Intent (POI) pool. Finally, the fine-tuned models are systematically compared with the original models using metrics such as execution rate, intent alignment, success rate, and BLEU.

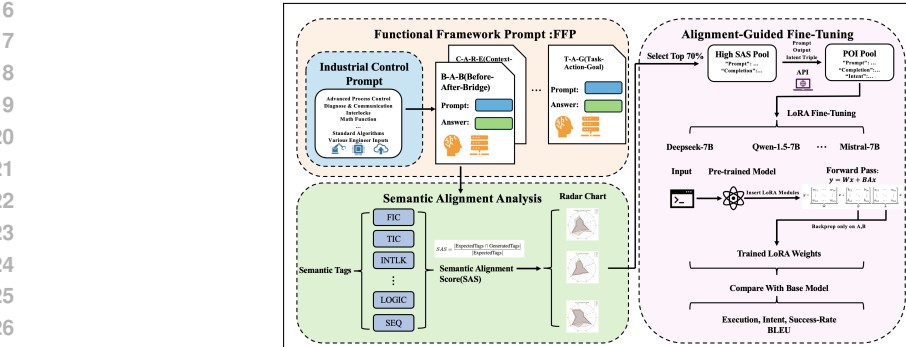

Figure 1: SAGE: semantic alignment analysis and alignment-guided fine-tuning with functional framework prompts

## 3.1 FFP: FUNCTIONAL FRAMEWORK PROMPT

We design five structured prompt formats to guide industrial control code generation. Each emphasizes a different reasoning aspect while remaining concise and interpretable. Due to space limitations, a detailed description is included in the Appendix.

### 3.1.1 BEFORE-AFTER-BRIDGE (B-A-B)

Emphasizes transformation from a baseline to an improved solution. **Example:** compare traditional PID control with a model predictive control implementation.

### 3.1.2 CONTEXT-ACTION-RESULT-EXAMPLE (C-A-R-E)

Links environment, action, and outcome to encourage descriptive reasoning. **Example:** reactor temperature control with an ANN replacing PID.

### 3.1.3 ROLE-INPUT-STEPS-EXPECTATION (R-I-S-E)

Highlights procedural reasoning and stepwise logic. **Example:** PLC startup sequence for a machine with heating, cooling, and feeders.

### 3.1.4 ROLE-TASK-FORMAT (R-T-F)

Maps task requests directly to a fixed-format output. **Example:** generate a PID controller as an IEC 61131-3 function block.

### 3.1.5 TASK-ACTION-GOAL (T-A-G)

Supports backward reasoning from goal to action, useful for diagnostics and optimization. **Example:** monitor PLC network health with watchdog logic.

## 3.2 SAS: SEMANTIC ALIGNMENT SCORE

### 3.2.1 CROSS-LINGUAL CONTROL SEMANTIC TAXONOMY

To ensure effective evaluation of functional correctness across industrial control code written in multiple languages—including Python, C++, MATLAB, and PLC Structured Text—we propose a language-independent control semantics taxonomy. This taxonomy covers key control intents commonly found in automation programs:

- **FIC: Flow Control** – Includes flow setpoint adjustment, flowmeter signal processing, and valve control to achieve stable flow.

- **TIC: Temperature Control** – Involves temperature measurement, PID regulation, heating/cooling commands, and thermal process optimization.
- **PIC: Pressure Control** – Covers pressure sensing, setpoint control loops, pressure relief valve logic, and pressure safety regulation.
- **INTLK: Interlock Logic** – Monitoring of safety-critical conditions, trip logic implementation, emergency stop, and permissive checking.
- **ALM: Alarm/Fault Handling** – Detection of abnormal conditions, alarm triggering, operator notification, and fault logging.
- **ACT: Actuator Commands** – Control signals for pumps, valves, motors, and other final control elements.
- **SEN: Sensor Acquisition** – Reading and conditioning process signals, scaling analog/digital inputs, and verifying sensor status.
- **LOGIC: Logical Structures** – IF/CASE decisions, Boolean operations, bitwise logic, and conditional control flow.
- **SEQ: Sequential Control** – Process step management, state machines, sequential function charts, and batch process sequencing.
- **RPT: Reporting and Logging** – Automatic report generation, event logging, trending, and historical data processing.

The taxonomy supports consistent semantic evaluation across PLC structured text, Python, and C++ platforms, facilitating the calculation of unified metrics such as the Semantic Alignment Score (SAS).

### 3.2.2 STRUCTURE RADAR CHART

We analyze how different prompt structures in the FFP framework affect the code generation corresponding to the aforementioned semantic labels. For each hint structure (e.g., R-I-S-E, C-A-R-E, T-A-G), we generate a set of code examples in multiple languages and label their control intent using a rule-based parser.

### 3.2.3 SEMANTIC ALIGNMENT SCORE

To quantify how well a cue achieves its target functionality, we define a semantic alignment score (SAS):

$$SAS = \frac{|\text{ExpectedTags} \cap \text{GeneratedTags}|}{|\text{ExpectedTags}|}$$

Expected tags are determined based on prompt intention (e.g., a prompt requesting "interlock logic" expects INTLK and ACT), while generated tags are extracted from code using multi-language semantic rules.Cross-language semantic alignment provides a robust foundation for evaluating the intent fulfillment of prompt-based code generation. By combining a unified control taxonomy with the SAS metric, we create a reusable benchmark for aligning LLM output with domain-specific control objectives.

### 3.3 AGFT:ALIGNMENT-GUIDED FINE-TUNING

To further improve the reliability and domain adaptability of large language models (LLMs) for industrial control code generation, we propose an alignment-guided fine-tuning (AGFT) method. This method leverages the alignment score between generated outputs and ground-truth structured text to guide sample selection and optimization during fine-tuning. The overall pipeline consists of three key components: selection of high-quality prompt-output-intent triples based on alignment metrics, lightweight fine-tuning using LoRA, and evaluation metric feedback.

### 3.3.1 FILTER AND ENHANCE USING INDUSTRIAL CONTROL SAMPLES

The proposed approach begins with an industrial control benchmark framework, which contains 1,500 diverse prompt-response pairs covering diverse industrial control domains. Each generated

code example is evaluated using a SAS, which quantifies how well the model-generated response matches the expected structure, logic, and format specifications. Examples with low SAS scores are filtered out, while those with high SAS scores are retained and further enhanced using data augmentation strategies (e.g., explaining the prompt, changing parameter constraints, or slightly adjusting the target condition) to enrich the fine-tuning framework.

### 3.3.2 LIGHTWEIGHT PARAMETER EFFICIENT FINE-TUNING

To reduce computational overhead, we employ Low Rank Adaptation (LoRA) as a fine-tuning method for the backbone network. LoRA inserts trainable rank-factorized weight matrices into pre-trained model layers, enabling efficient adaptation while freezing most of the original parameters. By applying LoRA to the decoder layers and attention heads, we are able to guide the generative behavior of the LLM towards better syntactic and semantic alignment without retraining the entire model.

### 3.3.3 PROMPT-OUTPUT-INTENT ENHANCEMENT

We introduced a three-way alignment strategy, constructing "prompt-output-intent" (POI) triplets. The "intent" element is a refined natural language summary automatically extracted from the original prompt (e.g., "Generate PID control code to maintain the reactor temperature at 180°C"). During fine-tuning, these POI triplets enable the model to understand both the prompt and the underlying goal. This additional context helps the model maintain fidelity to the control intent, especially in safety-critical or multi-stage processes.

### 3.3.4 EXPERIMENTAL PLAN

To implement AGFT in our experiments, we performed the following steps:

- **Step 1:** Compute the SAS score on an existing industrial control dataset using either a rule-based or learning-based alignment scorer.
- **Step 2:** Select the top 70% of high-SAS examples and augment them with smaller hint variants, tripling the sample pool.
- **Step 3:** Form point-of-interest triplets by automatically generating intents and injecting the intent tags as soft hints.
- **Step 4:** Apply LoRA-based fine-tuning on an open-source LLM (e.g., DeepSeek-7B or Qwen-1.5-7B) using HuggingFace's `peft` library.
- **Step 5:** Evaluate the fine-tuned model on the industrial control dataset using BLEU and Pass@k, and compare it to the baseline model.

This alignment-guided approach allows the model to selectively learn from highly aligned demonstrations, achieving improved performance while ensuring safety and compliance in industrial scenarios.

## 4 EXPERIMENT

### 4.1 EXPERIMENTAL SOFTWARE AND HARDWARE SETTINGS

The hardware and software setup for this experiment is as follows:
Hardware: Personal laptop, Alibaba Cloud A10 servers (4, each with 24GB of video memory)
Software: Vscode (SSH) (for model training and running basic code), Github (for storing datasets and results), HuggingFace (for downloading the model to be fine-tuned)

### 4.2 DATASET CONSTRUCTION DETAILS

The dataset is constructed based on 100 commonly used prompt words for industrial control scenarios. Based on these 100 prompt words, five different prompt methods (B-A-B, C-A-R-E, R-I-S-E, R-T-F, and T-A-G) were used to expand these 100 prompt words. These prompt words were then

Table 1: SAS distributions across models

(a) DeepSeek-7B

| Prompt | FIC | TIC | PIC | INTLK | ALM | ACT | SEN | LOGIC | SEQ | RPT |
|--------|-----|-----|-----|-------|-----|-----|-----|-------|-----|-----|
| B-A-B | 0.034 | 0.026 | 0.051 | 0.126 | 0.126 | 0.092 | 0.171 | 0.130 | 0.087 | 0.156 |
| C-A-R-E | 0.034 | 0.018 | 0.054 | 0.108 | 0.098 | 0.096 | 0.184 | 0.162 | 0.084 | 0.162 |
| R-I-S-E | 0.030 | 0.016 | 0.049 | 0.169 | 0.122 | 0.091 | 0.208 | 0.080 | 0.103 | 0.131 |
| R-T-F | 0.036 | 0.019 | 0.060 | 0.137 | 0.115 | 0.089 | 0.206 | 0.115 | 0.082 | 0.141 |
| T-A-G | 0.029 | 0.010 | 0.044 | 0.128 | 0.149 | 0.112 | 0.238 | 0.089 | 0.091 | 0.110 |

(b) GPT

| Prompt | FIC | TIC | PIC | INTLK | ALM | ACT | SEN | LOGIC | SEQ | RPT |
|--------|-----|-----|-----|-------|-----|-----|-----|-------|-----|-----|
| B-A-B | 0.034 | 0.016 | 0.053 | 0.135 | 0.119 | 0.111 | 0.208 | 0.103 | 0.077 | 0.145 |
| C-A-R-E | 0.024 | 0.011 | 0.048 | 0.149 | 0.125 | 0.093 | 0.215 | 0.106 | 0.106 | 0.125 |
| R-I-S-E | 0.031 | 0.010 | 0.048 | 0.176 | 0.130 | 0.099 | 0.216 | 0.097 | 0.081 | 0.112 |
| R-T-F | 0.037 | 0.011 | 0.053 | 0.130 | 0.122 | 0.101 | 0.202 | 0.114 | 0.085 | 0.144 |
| T-A-G | 0.032 | 0.012 | 0.050 | 0.136 | 0.136 | 0.099 | 0.191 | 0.117 | 0.089 | 0.136 |

(c) Grok

| Prompt | FIC | TIC | PIC | INTLK | ALM | ACT | SEN | LOGIC | SEQ | RPT |
|--------|-----|-----|-----|-------|-----|-----|-----|-------|-----|-----|
| B-A-B | 0.036 | 0.025 | 0.063 | 0.115 | 0.149 | 0.085 | 0.162 | 0.135 | 0.068 | 0.160 |
| C-A-R-E | 0.030 | 0.019 | 0.061 | 0.096 | 0.163 | 0.077 | 0.163 | 0.126 | 0.126 | 0.138 |
| R-I-S-E | 0.034 | 0.017 | 0.069 | 0.100 | 0.180 | 0.105 | 0.182 | 0.077 | 0.066 | 0.169 |
| R-T-F | 0.040 | 0.020 | 0.062 | 0.123 | 0.189 | 0.080 | 0.185 | 0.105 | 0.074 | 0.121 |
| T-A-G | 0.030 | 0.016 | 0.065 | 0.109 | 0.154 | 0.085 | 0.170 | 0.134 | 0.091 | 0.148 |

generated using three different mature large-scale model platforms (Grok, GPT, and Deepseek). Therefore, the dataset contains a total of 1,500 prompt words.

### 4.3 VISUALIZING SEMANTIC ALIGNMENT

Tables 1a, 1b, and 1c report the semantic alignment scores (SAS) distributions of different prompt structures on ten functional semantic labels for the Deepseek-7B, GPT, and Grok models, respectively. Overall, T-A-G consistently achieves the highest SAS on SEN labels across all models. For example, in Table 1a (Deepseek-7B), T-A-G scores 0.238 on SEN, significantly higher than B-A-B's 0.171, demonstrating its optimal performance in generating sensor-related logic. The R-I-S-E prompt performs exceptionally well on interlock (INTLK) and report (RPT) labels. For example, in Table 1b (GPT), R-I-S-E achieves 0.176 on INTLK, surpassing other prompt structures and demonstrating its effectiveness in generating protection and diagnostic logic. In contrast, the scores of C-A-R-E and R-T-F are more evenly distributed. For example, in Table 1c (Grok), C-A-R-E's scores across the ten labels show little variation, highlighting its generalization ability across different functional categories. Although the absolute scores vary between models, the relative performance trends of the prompt structures remain consistent. This result demonstrates the robustness and cross-model applicability of the Functional Prompt Format (FFP) dataset in guiding LLMs to achieve the target semantic intent. Furthermore, we use the radar chart in Figure 2 to visualize alignment trends, showing the SAS values for semantic labels for each prompt type across different models.

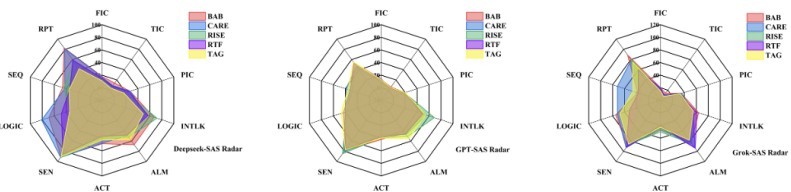

Figure 2: Radar chart of semantic alignment score

Table 2: Comparison of training losses and metric deltas across models

(a) Training losses over epochs

| Epoch | DeepSeek-7B | Gemma-7B | InternLM2-7B | Mistral-7B | Qwen2.5-7B |
|-------|-------------|----------|--------------|------------|------------|
| 0 | 1.49 | 2.45 | 2.22 | 0.91 | 1.92 |
| 1 | 0.99 | 2.14 | 2.03 | 0.62 | 1.08 |
| 2 | 0.94 | 2.41 | 1.74 | 0.51 | 0.72 |

(b) Pretrained vs. finetuned comparison on four metrics

| Model | $Exec_o$ | $Exec_t$ | $Intent_o$ | $Intent_t$ | $Succ_o$ | $Succ_t$ | $BLEU_o$ | $BLEU_t$ |
|-------|----------|----------|------------|------------|----------|----------|----------|----------|
| DeepSeek-7B | 0.676 | 0.837 | 1.000 | 1.000 | 0.424 | 0.601 | 3.45 | 6.62 |
| Gemma-7B | 0.647 | 0.625 | 1.000 | 1.000 | 0.389 | 0.358 | 3.31 | 2.96 |
| InternLM2-7B | 0.702 | 0.641 | 1.000 | 1.000 | 0.418 | 0.385 | 3.33 | 3.09 |
| Mistral-7B | 0.719 | 0.886 | 1.000 | 1.000 | 0.473 | 0.585 | 3.79 | 7.45 |
| Qwen2.5-7B | 0.716 | 0.653 | 1.000 | 1.000 | 0.467 | 0.379 | 4.06 | 3.04 |

## 4.4 ALIGNMENT-GUIDED FINE-TUNING LOSS RESULTS

Table 2a shows the training loss curves for five different 7B-scale language models during alignment-guided fine-tuning over three epochs. The loss of all models shows a downward trend, demonstrating that the models are able to effectively learn from the alignment-augmented dataset. Mistral-7B and Qwen2.5-7B show the most significant decreases in training loss, indicating better convergence. In contrast, Gemma-7B exhibits an unusual upward trend after the first epoch, suggesting possible overfitting or instability during fine-tuning. DeepSeek-7B and InternLM2-7B achieve higher final loss values than Mistral and Qwen, indicating slower convergence, but their performance steadily improves. These results highlight the variability in how models respond to alignment-guided supervised learning and emphasize the importance of architecture-specific tuning strategies.

Table 2b presents a comparative evaluation of the pre-trained and fine-tuned 7B-scale language models on four key metrics: Executability (Exec), Intent Consistency (Intent), Task Success (Success), and BLEU score. Among all models, Mistral-7B and DeepSeek-7B show significant improvements after fine-tuning, particularly in Executability (from 0.719 to 0.886 and 0.676 to 0.837, respectively) and BLEU score (from 3.79 to 7.45 and 3.45 to 6.62, respectively), indicating improved syntactic and semantic quality of the generated code. Notably, all models maintain perfect intent consistency (Intent = 1.000), confirming that the functional objective of the hint is preserved. However, models such as Gemma-7B and Qwen2.5-7B show performance degradation in terms of executableness, success rate, and BLEU after fine-tuning, suggesting possible overfitting or misalignment with the training distribution. These results indicate that while alignment-guided fine-tuning can significantly improve code generation performance, its effectiveness is model-dependent, highlighting the importance of architecture-aware fine-tuning strategies.

## 5 CONCLUSION

In this study, we introduce a structured and scalable pipeline for industrial control code generation by proposing three key innovations. First, we design a Function Prompt Format (FFP) dataset that enhances the controllability and domain-specificity of prompts through five well-defined structural formats. Second, we develop a semantic alignment score (SAS) mechanism based on intent label matching to quantitatively evaluate the degree of alignment between the prompt structure and the generated code semantics. Third, we propose an alignment-guided fine-tuning (AGFT) method that leverages SAS to guide data selection and improves the adaptability of the LLM through lightweight techniques such as LoRA. Our evaluation on multiple LLMs demonstrates consistent improvements in execution success rate, BLEU score, and functional intent preservation. These contributions not only enhance the interpretability and controllability of code generation in industrial automation but also highlight the feasibility of deploying domain-adaptive LLMs in safety-critical environments. Future work includes automatic prompt structure recommendation based on task intent, integration with retrieval-augmented generation (RAG), and the construction of larger industrial datasets for ongoing alignment and benchmarking.

# A APPENDIX

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

## B    DETAILED EXAMPLES OF FFP FORMATS

### B.1    BEFORE-AFTER-BRIDGE (B-A-B)

The B-A-B format is inspired by storytelling and writing templates that are effective in persuasive or comparative contexts. It includes:

- **Before:** A description of the traditional or current approach or challenge.
- **After:** A statement of the desired or improved future outcome.
- **Bridge:** A description of how the transformation will occur or how the model is expected to operate.

This structure is particularly effective when comparing baseline control logic with optimized solutions (e.g., traditional PID versus model predictive control). It guides LLMs in reasoning about the change process, which is critical for industrial control improvements.

**Example:**

- **Before:** In gas turbines, precise temperature control is critical for performance, safety, and emissions compliance. Turbine temperature fluctuations due to load or fuel variations can lead to inefficiency, overheating, or damage. Manual or poorly adjusted controls often fail to maintain stable turbine conditions.
- **After:** Develop an IEC 61131-3 structured text program that implements a PID loop to control turbine temperature, adjust the inlet valve to maintain the setpoint, adhere to valve limits, and ensure stable and responsive operation under varying conditions.
- **Bridge:** Develop IEC 61131-3 structured text code to calculate the error, update the PID term, and adjust the inlet valve position within a safe range to maintain the turbine temperature setpoint under varying loads.

## B.2 CONTEXT-ACTION-RESULT-EXAMPLE (C-A-R-E)

The C-A-R-E format is based on descriptive reasoning. Its structure is as follows:

- **Context:** The operating setting or physical environment.
- **Action:** A specific control action or method.
- **Result:** The expected or desired outcome.
- **Example:** A specific scenario or example of a requirement.

This prompt is well-suited for tasks that require interpreting physical processes as code or rules, particularly in batch control and safety logic scenarios.

**Example:**

- **Context:** In a chemical process, reactor temperature control is challenging due to its nonlinear and time-varying dynamics.
- **Action:** Develop a Python-based ANN controller and train it using historical data to predict and regulate reactor temperature.
- **Result:** Achieve stable temperature control with better adaptability and efficiency than a traditional PID controller.
- **Example:** Simulate a sudden change in feed temperature; demonstrate how the ANN can smoothly adjust to the overshoot and oscillation of the PID controller.

## B.3 ROLE-INPUT-STEPS-EXPECTATION (R-I-S-E)

The R-I-S-E format emphasizes procedural logic and helps guide the model through a defined workflow. Its components include:

- **Role:** The model's assumed role or responsibilities (e.g., controls engineer).
- **Input:** Available data, sensor signals, or initial conditions.
- **Steps:** The sequential actions or algorithms to be followed.
- **Expectation:** The final outcome or performance goal.

This format is particularly well-suited for sequential logic, state transitions, or batch operations.

**Example:**

- **Role:** You are a PLC programmer responsible for developing the startup logic for a 3D bag-making machine.
- **Input:** The machine contains eight heating stations, eight cooling stations, and dual feed rollers.
- **Steps:** Start the heating stations sequentially, enable the cooling system, and synchronize the feeder and cutter timing.
- **Expectation:** Use structured text to ensure safe and synchronized startup of all machine components.

## B.4 ROLE-TASK-FORMAT (R-T-F)

The R-T-F format is the most concise and direct. It is best used to quickly scope code generation tasks with fixed-format output. It includes:

- **Role:** The position or responsibility assumed by the model.
- **Task:** The specific action to be completed.
- **Format:** The desired output format (e.g., code, table, function block).

This format is often used in automated code generation platforms to enable rapid conversion from human language to code.

**Example:**

- **Role:** Play the role of a control system engineer.
- **Task:** Implement a PID controller in structured text to maintain the liquid level in a storage tank.
- **Format:** Output a complete IEC 61131-3 function block, including initialization and control logic.

### B.5 TASK-ACTION-GOAL (T-A-G)

The T-A-G format is useful when users need the model to reason backward from a goal and define its own path of action. Its structure is as follows:

- **Task:** General problem or high-level goal.
- **Action:** Proposed control strategy or approach.
- **Goal:** Operational outcome or process improvement.

T-A-G is well-suited for diagnostic code, optimization scenarios, and systems where the end result guides logic design.

**Example:**

- **Task:** Monitor the communication health of a PLC network.
- **Action:** Implement watchdog logic for OPC UA, Modbus, and Profinet.
- **Goal:** Ensure continuous diagnostics and generate alerts on connection failures.

## C  LLM USAGE DISCLOSURE

In preparing this work, large language models (LLMs) were used only as auxiliary tools. Specifically, LLMs assisted in (i) refining the English writing for clarity and grammar, (ii) checking LaTeX formatting and figure/table alignment, and (iii) generating small illustrative code snippets used as examples in the appendix. All core contributions of this paper, including the design of the Functional Framework Prompt (FFP) dataset, the development of the Semantic Alignment Score (SAS) metric, the proposal of the Alignment-Guided Fine-Tuning (AGFT) strategy, as well as dataset construction, experiments, and analysis, were entirely conceived, implemented, and validated by the authors. The authors take full responsibility for the content of this paper.

