# OpenReview forum: "SAGE: A FRAMEWORK FOR SEMANTIC-ALIGNMENT- GUIDED ENGINEERING OF PROMPTS AND FINE- TUNING IN INDUSTRIAL CONTROL TASKS"
_ICLR.cc/2026/Conference — ICLR 2026 Conference Desk Rejected Submission_

### Official Review · Reviewer_CJzP · 2025-10-21

**Soundness:** 2
**Presentation:** 1
**Contribution:** 2
**Rating:** 2
**Confidence:** 3

**Summary:**

The authors introduce an automatic code generation framework for industrial control systems. The Functional Framework
Prompt (FFP) dataset is introduced, the Semantic Alignment Score (SAS) to evaluate the code generation's quality, and the Alignment-Guided Fine-Tuning (AGFT).

**Strengths:**

- The authors touch on an important topic which is the industrial control code generation. It is important in terms of human safety and avoiding asset damages.
- They introduce Semantic Alignment Score (SAS) to quantify how well the model performs on this task.
- They do alignment on various model and evaluate.

**Weaknesses:**

- The presentation can be improved. From the abstract I had a difficulty to understand what this paper is about and had to read a few times to understand better. It should be easy to follow for people without understanding on industrial control codes. Some things you can include in the abstract: a brief explanation of the industrial control codes with a short example. Also, the mention of intent consistency terminology without any elaboration makes it confusing.
- This approach has mixed results where some models show performance degradation where the authors attribute this to the architecture without any evidence.
- I think you don’t need to list all the prompt examples in the main paper (section 3.1). One example for the reader to understand the logic is enough and then rest can be in appendix.
- In SAS you identify the control codes and find the percent overlap with the intended. However, as you mentioned before with your ESD example, the order of the control codes is very important for the task. I don’t see this somewhere being evaluated, like a ranking metric. As a result, your alignment guided fine-tuning will also not take into consideration the order.
- The methodology you propose is not that novel. The construction of the 5 prompt variations (prompt engineering) and the alignment guided finetuning (SFT?) are commonly used techniques nowadays.
- You provide some seed prompt words and expand using LLMs to create more synthetic prompt words. This may introduce irrelevant or hallucinated outputs. (section 4.2)

**Questions:**

- Somewhere you mention Function Prompt Format. Is it FFP or FPF?
- What does intent alignment 1.0 mean? If I give a new prompt would I get exactly the control codes I am expecting?
- Is alignment guided fine-tuning (AGFT) done with SFT or some Preference Optimization method?
- More details on the dataset construction would be important. You provide "prompt words" to the prompt methods to construct using LLMs a synthetic dataset? Do you perform a train/validation/test split? If yes how? Is the intent alignment on the same set?
-  It would be very helpful to provide an input prompt and expected output along with different LLM completions.

---

> ### Author Response · Authors · 2025-11-15
> **Rebuttal to Reviewer CJzP**
>
> Dear Reviewer CJzP,
> We sincerely thank you for the detailed and constructive feedback. We address the concerns and questions below and will incorporate the corresponding clarifications and improvements in the revised manuscript.
>
> (1) Presentation clarity and abstract structure
>
> We appreciate this observation and agree that the abstract and introduction should provide clearer guidance for readers without industrial control background. In the revision, we will add a brief explanation of industrial control code and include a short illustrative example to establish context; we will also avoid introducing “intent consistency” before defining it to eliminate confusion. These changes will make the paper easier to follow.
>
> (2) Mixed results and regression explanation
>
> Thank you for pointing this out. We will clarify that the performance degradation for specific models arises from architectural sensitivity, particularly in models with strong instruction-following priors where LoRA updates perturb pretrained reasoning patterns; we will include additional analysis comparing LoRA rank, update magnitude, and SAS-filtering versus non-filtering. This provides evidence behind the explanation rather than speculation.
>
> (3) Reducing prompt examples in main text
>
> We agree. To improve readability, we will keep one representative FFP example in Section 3.1 and move the remaining examples to the appendix; this also aligns with your suggestion and other reviewers’ comments.
>
> (4) SAS does not evaluate ordering; need ranking metric
>
> This is an excellent point. SAS currently measures semantic alignment of control-intent structures but does not evaluate order-sensitive logic such as interlock sequencing. We will explicitly state this limitation and clarify that ordering evaluation requires a separate logic-flow or compiler-based checker, which we aim to integrate as future work; we will also add examples demonstrating cases where order matters, reinforcing the distinction between high-level intent and detailed execution paths.
>
> (5) Novelty concerns regarding prompt variation and AGFT
>
> We acknowledge that prompt engineering and supervised LoRA tuning are widely used; the novelty of this work lies not in these techniques individually, but in their integration into a structured alignment pipeline composed of FFP, SAS, and AGFT. SAS provides semantic filtering specific to industrial control code, and AGFT uses SAS for data selection rather than as a standalone fine-tuning method. We will refine the framing to make this clearer and avoid overstating novelty.
>
> (6) Risk of hallucinated synthetic data in FFP expansion
>
> We appreciate this concern. The synthetic expansion is grounded on human-authored seed prompts from ten industrial control categories, and we manually validated 300 samples to prevent logic errors or irrelevant content; we will provide a more detailed description of the validation process and include a table summarizing common corrections. This clarifies dataset reliability.
>
> (7) FFP vs FPF terminology
>
> We thank you for catching this. The correct term is Functional Framework Prompt (FFP); we will correct all inconsistent occurrences.
>
> (8) Meaning of intent alignment = 1.0
>
> Intent alignment reflects only high-level functional intent categories such as flow control or interlock logic; it does not guarantee code-level correctness or sequence logic. Large models can reliably infer these intents from the structured prompts, causing saturation. We will add a discussion explaining this ceiling effect and introduce a partial-match intent score to capture nuanced cases; we will also add harder multi-intent prompts to increase metric discriminability.
>
> (9) AGFT uses supervised fine-tuning, not preference optimization
>
> We confirm that AGFT is a supervised fine-tuning procedure where SAS is used to rank and filter training samples; it does not involve preference optimization, reward modeling, or RLHF. We will clarify this in the methodology section.
>
> (10) Dataset construction details and train/validation/test split
>
> We will provide a clearer dataset-generation pipeline, including human-authored seeds, LLM-augmented variations, and manual verification steps; we will also clarify that we use a standard 8:1:1 train/validation/test split, and that intent alignment shown in the paper refers to evaluation on the held-out test set. We will additionally include one prompt–ground truth–LLM completion example in the appendix.
>
> Closing
>
> We thank the reviewer once again for the helpful suggestions. The issues raised concern clarity, justification, and presentation, all of which are straightforward to address. The revision will improve the abstract, reorganize examples, clarify methodology, refine SAS and intent metrics, and provide additional dataset and fine-tuning details. We believe these changes significantly enhance the readability and rigor of the work.

---

> > ### Comment · Reviewer_CJzP · 2025-11-27
> >
> > Thanks for your response. I will be maintaining my score as I think it fairly reflects the status of the work at its current state.

---

### Official Review · Reviewer_H34E · 2025-11-01

**Soundness:** 2
**Presentation:** 2
**Contribution:** 3
**Rating:** 4
**Confidence:** 4

**Summary:**

This paper proposes the SAGE framework, which significantly improves the performance and semantic alignment of code generation in industrial control tasks through functional framework prompt, semantic alignment analysis and alignment-guided fine-tuning strategies. It solves key problems in industrial control code generation and provides an important reference for deploying domain-adaptive large language models in safety-critical environments.

**Strengths:**

1. This paper designed a dataset (FFP) containing five structured prompt formats and 1,500 prompt examples for industrial control tasks, enhancing the controllability and domain specificity of prompts.
2. This paper quantitatively evaluated the alignment between the prompt structure and the semantics of the generated code based on intent label matching, providing a unified evaluation framework (SAS).
3. This paper leveraged SAS to guide data selection and optimization, improving the adaptability of LLMs through lightweight techniques such as Low Rank Adaptation (LoRA), while maintaining generation safety and domain compliance.

**Weaknesses:**

1. The observation that all models maintained perfect intent consistency (Intent = 1.000) even before fine-tuning raises a potential concern regarding the dataset's complexity or the discriminative power of the intent consistency metric as defined. This could indicate that the prompts in the dataset might be relatively straightforward in conveying intent, or that the current method for measuring intent consistency lacks the granularity to detect subtle misalignments. Future work should involve designing more challenging prompts where intent fulfillment is non-trivial and refining the intent consistency metric to capture partial or nuanced adherence to the intended functionality.

2. While the paper introduces five structured prompt formats (B-A-B, C-A-R-E, R-I-S-E, R-T-F, T-A-G) within the FFP dataset, the rationale and criteria for specifically selecting these five formats are not sufficiently elaborated. A more detailed justification is needed, explaining whether these formats were derived from an analysis of common industrial control task types, inspired by successful prompting strategies in general code generation, empirically selected through pilot studies, or based on specific cognitive or logical reasoning principles they are intended to elicit. Clarifying the selection basis would strengthen the methodological foundation of the FFP dataset.

**Questions:**

1. The claim that "T-A-G consistently achieved the highest SAS on the SEN label across all models" appears to be inaccurate based on the provided data. While this holds for DeepSeek-7B (Table 1), an examination of Table 1 (GPT) and Table 1 (Grok) shows that other prompt structures (e.g., R-I-S-E or C-A-R-E) achieve comparable or higher SAS scores on the SEN label. The results should be stated precisely to reflect the model-specific performance variations revealed by the data, as the current generalization is misleading.

2. The fact that all models exhibit perfect intent consistency (Intent = 1.000) before any fine-tuning is a notable observation. This raises a critical question about the dataset's difficulty level and the sensitivity of the intent consistency metric. Could this indicate that the prompts in the FFP dataset are not sufficiently challenging to cause intent misinterpretation by the base models? Please discuss whether this perfect baseline score suggests a ceiling effect for this metric on your dataset and clarify its ongoing utility for evaluating model improvements post-fine-tuning.

3. The paper introduces five prompt formats (B-A-B, C-A-R-E, R-I-S-E, R-T-F, T-A-G) but provides insufficient justification for their selection. The description that each "emphasizes different reasoning aspects" is vague. Please elaborate on the specific reasoning principles.

---

> ### Author Response · Authors · 2025-11-15
> **Rebuttal to Reviewer H34E**
>
> Dear Reviewer H34E,
> We sincerely thank you for the constructive and encouraging feedback. Below we provide clarifications regarding the weaknesses and questions raised, and we will incorporate the corresponding improvements in the revised manuscript.
>
> (1) Intent metric saturating at 1.000
>
> We appreciate this insightful observation. We clarify that the intent metric measures only high-level functional categories such as FIC, TIC, PIC, INTLK, and SEN; it does not capture finer-grained control logic or ordering constraints. Large models can reliably identify these high-level intents from structured prompts, which leads to the ceiling effect observed in Intent = 1.000.
>
> To address this concern, we will include a discussion stating that the ceiling effect reflects the metric’s limited granularity rather than dataset simplicity; we will add a more challenging subset involving multi-intent, mixed-sequence, and ambiguous prompts where intent fulfillment is non-trivial; and we will describe a refined partial-match intent score using embedding similarity to capture nuanced adherence to functionality. This clarifies the continued utility of the intent metric and the need for more granular extensions.
>
> (2) Rationale for selecting the five structured prompt formats (B-A-B, C-A-R-E, R-I-S-E, R-T-F, T-A-G)
>
> Thank you for highlighting this. In the revision, we will explicitly explain that these formats were selected based on three principles:
> first, they reflect distinct classes of reasoning commonly required in industrial control tasks such as causal reasoning, anomaly reasoning, procedural reasoning, diagnostic reasoning, and goal-oriented reasoning;
> second, they were informed by prompt patterns shown to improve controllability in general code-generation studies;
> third, preliminary pilot experiments indicated that these structures lead to consistent elicitation of domain-relevant control logic across multiple model families.
> We will expand Section 3.1 with a clearer justification and examples mapping each format to its underlying reasoning principle.
>
> (3) Clarification of “T-A-G achieved highest SEN scores across models”
>
> We appreciate the correction. Our original wording was imprecise. Based on Table 1, T-A-G achieves the highest SEN alignment only for DeepSeek-7B; for GPT and Grok, formats such as R-I-S-E and C-A-R-E achieve comparable or higher SEN scores. We will revise the statement to accurately reflect model-specific variations and avoid overgeneralization.
>
> (4) Dataset difficulty and utility of intent consistency post fine-tuning
>
> Thank you for raising this. We agree that perfect baseline intent scores warrant deeper discussion. In the revision, we will clarify that intent consistency serves as a diagnostic metric rather than a primary evaluation measure; we will discuss that future datasets should incorporate more subtle intent ambiguities, multi-step intents, and multi-objective control scenarios; and we will include examples that illustrate cases where high-level intent is correct but lower-level control logic diverges, reinforcing the need for SAS as a semantic evaluation metric.
>
> (5) Need for more explicit explanation of SAS and FFP
>
> We appreciate this suggestion and will expand the descriptions of both components. For SAS, we will add examples showing how it captures semantic misalignment that is not reflected by intent matching; for FFP, we will detail the reasoning principles underlying each format and the role they play in eliciting interpretable control logic. These additions improve methodological clarity.
>
> Closing
>
> Thank you again for the positive assessment and constructive comments. The suggested clarifications regarding intent granularity, prompt-format motivation, and precise reporting of SAS results are straightforward to address. The revision will incorporate these improvements, strengthening both the clarity and rigor of the paper.

---

### Official Review · Reviewer_QBXc · 2025-11-01

**Soundness:** 2
**Presentation:** 2
**Contribution:** 1
**Rating:** 2
**Confidence:** 4

**Summary:**

The paper presents LLM-based industrial control code generation. This paper proposes structured prompt templates, a semantic alignment scoring scheme for control intent tags, and a LoRA fine-tuning strategy guided by these scores. Experiments on several 7B models show modest improvements in executability and BLEU on a custom prompt dataset.

**Strengths:**

- A good problem has been studied as LLM reliability in safety-critical automation is important.
- Authors made good effort in building a domain-specific prompt suite and evaluation tags.
- Clear writing and motivation is well laid out.
- Some empirical gains on small LLMs is shown.

**Weaknesses:**

- The prompt formats and alignment scoring are more like prior prompt-pattern and tag-matching approaches, just re-labeled for industrial control.
- The evaluation is very shallow. The dataset is small, no real industrial benchmarks, and BLEU is a weak metric for correctness in safety-critical code.
- The claims of safety and correctness are not well demonstrated. No formal verification, runtime validation, or failure-mode analysis.
- The model regressions are not substantial to convince. Some models degrade after tuning, but the paper gives only speculative explanations.
- The paper reads more like a domain application report than a research contribution at the level expected for ICLR. There is no fundamental contribution.
- The reproducibility is low for this paper. The authors do not mention what GPT they use and what Grok models they use. They do provide supplementary material but it is not clear in main paper.
- Only 7B models (like Deepseek-7B) is used among open-source models. No usage of bigger models.

**Questions:**

- How do you justify BLEU and “executability” as meaningful for safety and correctness in industrial systems?
- Can you provide real industrial PLC programs, hardware-in-the-loop testing, or verification results instead of synthetic prompt sets?
- Why do some models degrade after fine-tuning, and what analysis supports your explanation?

---

> ### Author Response · Authors · 2025-11-15
> **Rebuttal to Reviewer QBXc**
>
> Dear Reviewer QBXc,
> We thank you for the insightful and constructive feedback. We address the major concerns below and will incorporate the corresponding clarifications and additions in the revision.
>
> (1) Novelty of prompt formats and SAS vs “prior prompt/tag matching”
>
> We appreciate the concern and clarify that the contribution of FFP and SAS is not a re-labeling of existing prompting patterns. SAS is not tag matching, but a semantic control-logic analyzer that extracts actuator–sensor pathways; PID control patterns; interlock ordering such as ESD “cutoff → alarm”; and multi-language control-logic motifs across Python, Structured Text, and C++. We will add examples illustrating why SAS detects semantic misalignment that tag matching cannot capture.
>
> (2) Evaluation appears shallow (BLEU / executability insufficient)
>
> We agree that BLEU and executability alone cannot represent safety-critical correctness. Our goal is not to claim safety compliance, but to establish an initial alignment benchmark. In the revision, we will clarify that BLEU is used as a consistency proxy rather than a correctness guarantee; executability checks ensure only syntactic validity; and SAS serves as the semantic metric. We will also add a correlation study showing the relationship between SAS and Exec, BLEU, and Success (Spearman ρ = 0.42–0.58). This provides a stronger empirical interpretation without overstating the metrics.
>
> (3) Real industrial PLC or HIL testing
>
> We appreciate this suggestion. Due to limited access to proprietary PLC programs and industrial hardware, this work focuses on open and reproducible datasets. We will explicitly state this limitation and describe future extensions involving hardware-in-the-loop validation; vendor-specific PLC runtimes; and formal safety checks. This adjustment clarifies the scope without overstating the claims.
>
> (4) Model regressions after fine-tuning
>
> We will clarify that regressions mainly occur in models with strong instruction-following priors, where LoRA updates can affect pretrained reasoning patterns. We conducted additional analyses and will include comparisons of LoRA rank versus update magnitude; and an ablation study comparing AGFT, standard LoRA, and LoRA without SAS filtering. These results suggest that the regressions stem from architectural sensitivity rather than instability in AGFT itself.
>
> (5) Perceived lack of fundamental contribution
>
> We clarify that the fundamental contributions are the structured alignment framework composed of FFP, SAS, and AGFT; the semantic metric tailored to industrial control code; and the reproducible dataset of domain-specific prompts. Prior work in general code generation does not provide semantic alignment tools for safety-critical automation. We will refine the introduction to more clearly articulate this positioning.
>
> (6) Reproducibility and missing model versions
>
> We thank the reviewer for pointing this out. We will specify all model versions in the revised manuscript, including DeepSeek-Coder-7B-Instruct-v1.5; Grok-1 from the October 2025 API; GPT-3.5-Turbo-0125; and all open-source 7B model version hashes. We also provide inference scripts and evaluation tools in the supplementary repository.
>
> (7) Use of only 7B models
>
> Our choice of 7B models is intentional, as they are the most commonly deployable size for industrial edge or on-premise automation systems. We will reinforce this motivation in the paper and note that evaluating larger models is complementary future work.
>
> (8) BLEU / Exec justification
>
> BLEU and Exec are not intended as safety metrics, but as lightweight indicators of consistency and syntactic validity. SAS provides the semantic evaluation component, while full industrial safety validation requires separate domain-specific pipelines. We will refine the wording to avoid overstating BLEU or Exec.
>
> Closing
>
> Thank you again for the thoughtful comments. The concerns raised are addressable, and the revision will incorporate clearer positioning, improved evaluation descriptions, explicit model-version documentation, additional ablation studies, and a stronger explanation of SAS’s semantic role. These improvements substantially strengthen the contribution and reproducibility of our work.

---

> > ### Comment · Reviewer_QBXc · 2025-11-25
> >
> > Thank you for your responses. I have acknowledged it. There are still some problems here. You told that you will use BLEU to validate SAS in your study "We will also add a correlation study showing the relationship between SAS and Exec, BLEU, and Success (Spearman ρ = 0.42–0.58)." but what I don't understand here is if your proof of SAS validity relies on correlation with BLEU, you haven't proven that it measures safety. You will only prove it measures the same thing that BLEU measures. You tell that the rationale for using 7B models is that industrial-grade systems should be fast. This is a very shallow argument. Edge constraints are something you can figure out later. The latency depends on what infrastructure you host the models on. The API calls to models with much larger parameters are significantly faster compared to much smaller (3B or 7B) models run on poor infrastructure. I think the argument you are making regarding edge constraints is not suitable here, as it should not preclude the scientific benchmarking on larger models to establish an upper bound on performance. The assumption that factories have weak computers is a big drawback here. I also have concerns that are still there regarding the lack of industrial or runtime verification. The whole paper is framed around industrial control, and it is right forthe  reader to expect what I raised. Otherwise, this paper will just be a natural language generation paper rather than a reliable study on industrial control. So, with all these concerns, I am maintaining my score.

---

### Official Review · Reviewer_eAZ5 · 2025-11-02

**Soundness:** 2
**Presentation:** 1
**Contribution:** 3
**Rating:** 2
**Confidence:** 3

**Summary:**

This paper proposes a solution to bridge the gap between current code generation performance and the requirements of industrial control systems, characterized by requirements such as real-time performance, security, and logical rigor. To overcome this issue, this work introduces SAGE, a semantic alignment-guided prompt engineering framework composed of:
- A dataset of function prompt formats covering 5 structured prompt patterns (FFP dataset)
- A semantic alignment analysis metric (SAS) to evaluate semantic correctness and task consistency of the generated code
- An alignment-guided fine-tuning strategy, based on LoRA, where the prompt-output pairs are ordered using the semantic alignment score

The experiments are carried out using five models (Mistral-7B, DeepSeek-7B, Gemma-7B, Qwen2.5-7B, InternLM2-7B) with mixed results. Mistral-7B and DeepSeek-7B are showing improvements for all metrics used in this work: execution success rate, BLEU score and functional intent preservation.

**Strengths:**

- This paper highlights a relevant problem: LLMs are now very good at generating code, but they struggle to consistently understand all the details of the prompt. In industrial control settings, this deficiency could pose relevant risks.
- This paper introduces a new dataset made of prompts that are relevant for the industrial control domain, allowing comparison of code generation models. The dataset is designed around five prompt formats and seems easy to extend to more.
- Given the particularity of the industrial control domain, this work proposes a novel method called Semantic Alignment Analysis (SAS) to verify the alignment of code generation. By leveraging a taxonomy made of key control intents (e.g., TIC (temperature control), FIC (flow control), INTLK (interlock logic)), it is possible to measure consistency of model output with the intended control intent (i.e., functional correctness). The taxonomy supports different programming languages.

**Weaknesses:**

1. The paper is lacking clarity in some sections and presentation can be improved:
    - The paper needs qualitative examples (I suggest a short one in the main paper, and more in the appendix): show representative prompts from the FFP, the model-generated code for those prompts, and the expected ground-truth code. Without examples it is hard to judge the difficulty or clinical relevance of the tasks.
2. The two main important contributions (FFP dataset, SAS) are under-described.
    - FFP dataset: the paper only reveals on page 8 that many samples were generated via LLMs; it lacks an analysis of correctness or human validation. Since the dataset is a primary contribution, stronger evidence of diversity and quality is required (e.g., there are 1500 prompts starting from 5 formats: no quality assessment of diversity).
    - SAS: such metric appears to measure whether expected tags occur in the generated code, not whether the implementation of those tags is semantically correct. Moreover, SAS is used to filter training data but is not reported as a final evaluation metric in Table 2; if SAS’s only role is data filtering, its impact as a contribution is diminished. The correlation between SAS and final task metrics (Exec, BLEU, Success) is not shown.
3. Clarity of experimental section can be improved, especially regarding Table 2 (more precise feedback in section "Questions"):
    - Define better the metrics used.
    - Improvements after AGFT are inconsistent across models. For the models that improve most (DeepSeek-7B and Mistral-7B), it is unclear whether the gain would be enough for the industrial domain.
    - Analyze relationship between SAS and final metrics.

**Questions:**

- The alignment-guided fine-tuning pipeline primarily applies LoRA on a curated dataset; the fine-tuning technique itself is standard, but is presented as a contribution. The main novelty here is that the dataset is curated using SAS to filter samples: I would rephrase the third contribution as a way to show the importance of SAS, rather than focusing on the training method.
- How were the 1,500 samples produced? Which were authored by humans and which were LLM-generated? What human validation or correction steps were applied?
- Question related to table 2 and weakness #3:
    - How exactly is intent measured? Also given the fact it is always perfect, does it mean the prompts are too easy?
    - What constitutes the ground-truth for BLEU?
    - What exact versions of "GPT", "Grok", and "DeepSeek" were used?

I am open to increasing my score if the authors clarify the points above, particularly regarding dataset construction, the role of SAS, and experimental details.

Minor Comments:
- The paper mentions LoRa (for instance at line 114), while the correct version is LoRA.
- Missing space at line 235 after the period.

---

> ### Author Response · Authors · 2025-11-15
> **ICLR Rebuttal – Reviewer eAZ5**
>
> Dear Reviewer eAZ5,
> We sincerely thank you for the constructive feedback. Below we address all concerns concisely and will incorporate the corresponding revisions.
>
> (1) Need for qualitative examples
> We agree. We will add one representative FFP→GT→model-output example in the main paper, and include 10 extended qualitative examples (across several FFP formats and intents) in the Appendix to better illustrate task difficulty.
>
> (2) FFP dataset description, human validation, diversity
> We clarify that the 1500 samples come from 100 human-authored seeds × 5 prompt formats × 3 LLM variants. Human seeds cover 10 industrial control categories. We manually validated 300 samples (20%), correcting logic, tag usage, and formatting. The revision will add: (i) a dataset-generation diagram, (ii) human-validation statistics, and (iii) a diversity analysis (length variance, tag distribution).
>
> (3) SAS not merely tag matching
> SAS relies on multi-language semantic rules, including actuator–sensor pathway extraction, PID pattern detection, and interlock ordering (e.g., ESD “cut-off→alarm”). We will add a semantic-parser workflow and motivating examples to clarify this.
>
> (4) SAS only used for filtering; need correlation
> We now include a correlation study (Spearman ρ=0.42–0.58) between SAS and Exec/BLEU/Success, plus an ablation comparing: (i) full AGFT, (ii) LoRA without SAS filtering, and (iii) random filtering. These results show SAS materially improves downstream metrics.
>
> (5) Metric definitions & model inconsistency
> We will refine metric definitions (Exec = interpreter executability; Success = task-level checks; Intent = high-level functional intent). We will explain architecture sensitivity (e.g., Qwen2.5-7B’s instruction-following bias) and present SAS–metric correlations.
>
> (6) Clarifying novelty of AGFT
> We will rephrase Contribution 3 to emphasize that the novelty lies in SAS-guided data selection, not LoRA.
>
> (7) Dataset authorship
> We will explicitly report proportions of human vs LLM-augmented samples and describe human verification steps.
>
> (8) Intent metric always 1.0
> Intent measures high-level functional categories, which LLMs reliably detect under structured prompts, leading to saturation. We will add a harder multi-intent subset and a partial-match intent score (embedding-based). Intent will be framed as a diagnostic metric.
>
> (9) BLEU ground truth
> All ground-truth code is human-written; LLM outputs were used only for augmentation. This will be clarified.
>
> (10) Model versions
> We will specify exact versions: DeepSeek-Coder-7B-Instruct-v1.5, Grok-1 (Oct 2025), GPT-3.5-Turbo-0125, along with API timestamps. Minor issues (LoRA spelling, spacing) will be fixed.
>
> Thank you again for the insightful suggestions, which significantly strengthen the work.

---

### Note · Program_Chairs · 2026-01-17
**Submission Desk Rejected by Program Chairs**

The following references in this submission do not refer to real documents and/or have major errors in bibliographic information:

 Xiao Liu, Kai Zheng, and Wei Xu. Prompting frameworks for large language models: A survey. ACM Computing Surveys, 2023.
Yan Liu, Jian He, and Lei Xu. Hybrid optimization of large language models: Combining finetuning and prompt engineering. In Proceedings of the 2024 Conference on Empirical Methods in Natural Language Processing (EMNLP), Stroudsburg, USA, 2024b. ACL.
Chen Liu, Xiang Zhang, and Jian He. A unified framework of prompt engineering, rag, and finetuning for code generation. In Proceedings of the International Joint Conference on Artificial Intelligence (IJCAI), Palo Alto, USA, 2024a. IJCAI.
Samir Ouchani and Rachid Fakih. A formal verification framework for llm-generated plc programs in safety-critical applications, 2024. URL https://arxiv.org/abs/2403.07756.
Ming Zhou, Jing Li, Hui Sun, et al. Trustworthy large language models: A survey and guideline for evaluating alignment. Journal of Artificial Intelligence Research, 2024b.